# LieRE: Lie Rotational Positional Encodings

**Sophie Ostmeier** [1] [*]  **Brian Axelrod** [*]  **Maya Varma** [1]  **Michael Moseley** [2]  **Akshay Chaudhari** [2] [†]  **Curtis Langlotz** [2] [†]

## Abstract

Transformer architectures rely on position encodings to model the spatial structure of input data. Rotary Position Encoding (RoPE) is a widely used method in language models that encodes relative positions through fixed, block-diagonal, rotation matrices applied to key-query interactions. We hypothesize that this inductive bias limits their RoPE's effectiveness for modalities with high dimensional structure. Lie Relative Encodings (LieRE) introduce a principled generalization of RoPE, aimed at increasing the representational capacity of positional encodings in transformers. Instead of fixed 2D rotations, LieRE learns dense skew-symmetric matrices (Lie algebra elements), which are then differentiable mapped to form high-dimensional rotation matrices (Lie group elements). This results in richer, learnable, and continuous, encodings of both relative and absolute positional information. We demonstrate the effectiveness of LieRE on 2D and 3D vision tasks, showing that it generalizes well to higher input resolutions while maintaining computational efficiency. The code and checkpoints are publicly available at https://github.com/StanfordMIMI/LieRE.

## 1. Introduction

The attention mechanism, particularly within transformer architectures, has revolutionized machine learning across diverse domains. However, attention is inherently permutation invariant—it cannot leverage the sequential order of its inputs as information directly. This fundamental limitation necessitates additional mechanisms to encode positional information, enabling models to capture sequential and spatial dependencies crucial for tasks ranging from natural language to image understanding (Vaswani et al., 2017).

This challenge has sparked extensive research into positional encodings, which inject order information into the otherwise order-agnostic attention mechanism. The field has evolved from early approaches using fixed sinusoidal positional embeddings (Vaswani et al., 2017) to more sophisticated learned embeddings (Shaw et al., 2018a; Devlin et al., 2019; Dosovitskiy et al., 2020). Recent advances have introduced increasingly dynamic methods, including relative position representations (Shaw et al., 2018b; Dosovitskiy et al., 2020) and rotary positional embeddings (Su et al., 2024; Heo et al., 2024), demonstrating the critical role of position encoding in enabling transformers to effectively process ordered sequences.

In particular, Rotary Position Encoding (RoPE) has emerged as an elegant solution for encoding relative positional information in transformers (Su et al., 2024). RoPE works by applying a rotation matrix to each token's keys and queries, where the rotation angle depends on the token's absolute position in the sequence. The key insight is that when two tokens interact through the attention inner product, their rotated representations naturally encode their relative distance. For example, when a token at position five attends to a token at position two, RoPE enables the model to understand that these tokens are three positions apart, regardless of their absolute positions in the sequence. This translation-invariant property makes RoPE particularly efficient at capturing position-dependent patterns in text, which has led to its adoption in popular open-source language models such as LLaMA and Mixtral.

Despite RoPE's success in sequential tasks (Touvron et al., 2023; Chowdhery et al., 2023), it faces several fundamental limitations. First, RoPE is designed for one-dimensional sequence data with an exponential decay in frequencies, making it suboptimal for spatial relationships in images or spatiotemporal patterns in videos where distant pixels may have strong correlations. Second, when handling higher dimensional data, the challenge compounds significantly– learned relative position encodings must capture exponentially many relative positions for $n$-dimensional data, as demonstrated in 2D image data where positions must be encoded both horizontally and vertically (Shaw et al., 2018a).

---

[*]Equal contribution  [1]Computer Science Department, Stanford University, USA  [2]Radiology Department, Stanford University, USA. Correspondence to: Sophie Ostmeier <sostm@stanford.edu>, Brian Axelrod <baxelrodresearch@gmail.com>. [†] Equal senior authorship.

*Proceedings of the $42^{nd}$ International Conference on Machine Learning*, Vancouver, Canada. PMLR 267, 2025. Copyright 2025 by the author(s).

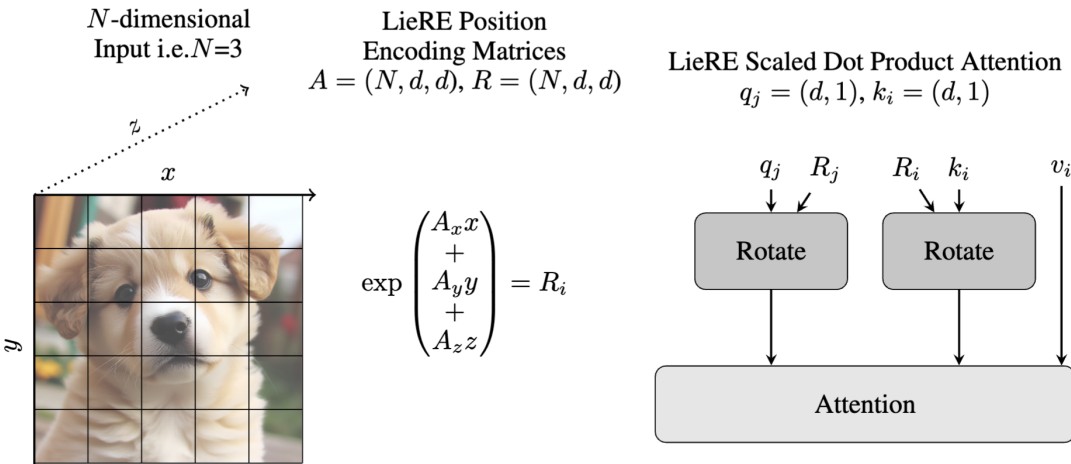

*Figure 1.* Overview of how LieRE encodes spatial information, where $N$ denotes the number of input dimensions, i.e. $N = 3$ for the 3D image. $A$ is a learnable skew symmetric matrix and $R_i = \exp(A[x_i \ y_i \ z_i]) \in \mathbb{R}^{d \times d}$ is the rotation matrix for the $i$th patch in the flattened input, where $j$ corresponds to a different patch. $d$ is the head dimension.

Third, RoPE's reliance on sparse, block-diagonal rotation matrices with handcrafted basis functions constrains its ability to learn complex spatial dependencies (Chu et al., 2024).

Our key insight is that Lie group theory provides a natural framework for generalizing relative position encodings to higher dimensions. We introduce Lie Relative Encodings (LieRE), which replaces RoPE's rotation matrices with learned, dense rotations derived from Lie group generators. LieRE learns a basis of skew-symmetric matrices $A_i$, computes rotation matrices $R(p) = \exp(\sum_{i=0}^{n} p_i A_i)$ for $n$-dimensional positions, and applies these rotations to keys and queries in the attention mechanism. The relative positions are then naturally captured through the inner product of the rotated keys and queries.

This approach addresses RoPE's limitations in two key ways: **(1) it handles higher-dimensional spatial relationships through Lie groups** and **(2) it increases representational capacity through variably dense, learned rotation matrices** while requiring only 0.68% of parameters in a ViT-B model. Importantly, LieRE can be implemented with a single modification to existing architectures.

To assess the impact of LieRE and other positional encodings on ViT performance, we evaluate several encoding schemes across diverse tasks, including 2D and 3D image classification. Additionally, we investigate a fundamental spatial reasoning task where models must identify where an arrow points. Despite their sophistication, contemporary multimodal LLMs struggle with this seemingly simple task. Our experiments reveal that successful completion of this task specifically requires relative position encodings,

highlighting their crucial role in spatial understanding.

## 2. Related Work

### 2.1. Position Encodings

We split the review of positional encodings into: a) absolute, b) relative, and c) contextual.

*Absolute* position in this context refers to a position with respect to a consistent reference, usually the start of a text or top left corner of an image. Absolute encodings generally operate on a per token-level, modifying the embedding of a token to encode the location of the token in the input. Methods such as sinusoidal and learned absolute encodings directly add vectors to the input token embedding (Vaswani et al., 2017; Devlin et al., 2019; Dosovitskiy et al., 2020).

*Relative* position encodings, in contrast, encode the relative positions of two tokens. One strategy is to learn an set of embeddings for position deltas which can be incorporated into the attention mechanism (Shaw et al., 2018b; Liu et al., 2021; 2022). However, this incurs quadratic computational cost in terms of the number of tokens as a separate embedding is required for every pair of tokens. Rotary Position Encodings (RoPE) avoid this cost by rotating the key and query vectors before the attention inner product. The algebraic properties of the block-diagonal rotation matrices used in RoPE ensures only relative positional information is captured in the attention mechanism (Su et al., 2024). RoPE is quite widely used in open source LLMs including the PaLM, Llama and Mixtral models (Touvron et al., 2023; Chowdhery et al., 2023; Jiang et al., 2024). However, RoPE

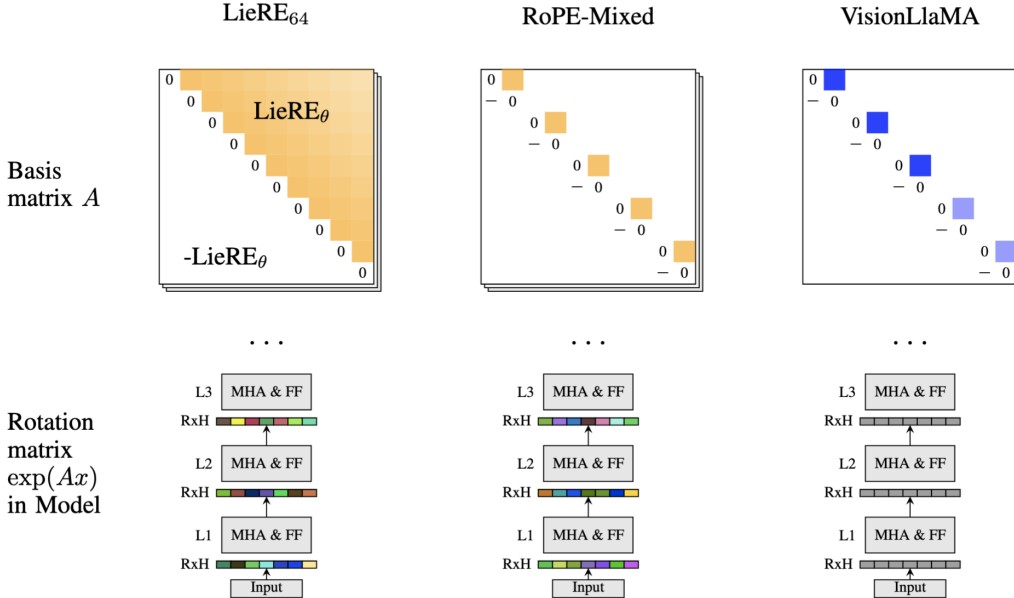

*Figure 2.* Comparisons of basis matrix $A$ of our work, Rope-Mixed (Heo et al., 2024), VisionLlama (Chu et al., 2024). First row: learnable parameter (yellow), not learnable parameter (blue) in basis matrix. Second row: rotation matrices $\exp(Ax)$ shared across the stem (gray) or different for each layer and head (colorful). MHA := Multi-Head-Attention, FF := Feed Forward, L[index] := Layer index, RxH := Rotation Matrix per Head, $LieRE_{\theta}$ := the number of LieRE parameters.

can perform poorly on inference for larger context sizes than the model was trained on. This has spurred an active line of work extending RoPE to longer contexts, work which we review later.

We refer to the last category of positional encodings as *contextual* position embeddings. This category is defined by encodings that aim to capture semantic positional information lost in traditional absolute and relative position encodings, often motivated by reasoning or mathematical tasks. Contextual Position encodings achieve (CoPE) this by allowing the model to learn how the position is computed (Golovneva et al., 2024). Abacus embeddings enable transformers to learn how to handle arithmetic by better exposing the digit structure of numbers (McLeish et al., 2024).

## 2.2. Extensions of RoPE

The efficiency and popularity of RoPE have led to several lines of work building off of it.

One notable one is context extension, which aims to address the fact that RoPE NLP models trained on short documents tend to perform poorly on long documents. Methods like NTK-aware context extension, YaRN and LongRoPE focus on enabling already trained models to handle long context, both with and without finetuning (Ding et al., 2024; Peng et al., 2023; Tworkowski et al., 2024; Chen et al., 2023).

The final, and most relevant, line of work has been specifi-

cally focused on adapting RoPE to image tasks. Both Vision-Llama and RoPE-Mixed present relative position encodings inspired by RoPE capable of encoding 2D positional data (Chu et al., 2024; Heo et al., 2024). The primary difference is that RoPE-Mixed has a learnable component, whereas VisionLlama does not.

## 2.3. Efficient Scaling beyond Sequence Data

There is also an extensive line of work improving the performance of transformers for modalities with dimensionality greater than one. Axial Attention (Ho et al., 2019) reduces computational complexity by applying attention along specific axes (e.g., rows and columns in images), enabling transformers to scale efficiently with high-dimensional data. Perceiver utilizes latent tokens to compress high-dimensional inputs into a smaller set, improving scalability as input size and dimensionality grow. These methods address the inefficiencies of traditional transformers when applied to high-dimensional data (Jaegle et al., 2021). Additionally, techniques like Swin and Vmamba optimize compute for visual data through structuring how information flows through the network (Liu et al., 2021; 2024). Swin Transformer introduces a hierarchical approach with shifted windows, limiting attention to local regions to reduce complexity while capturing global context. Vmamba, on the other hand, proposes a visual state space model that represents images as a collection of spatial states, allowing attention to be applied

efficiently across large-scale visual inputs by exploiting spatial locality and reducing redundant computation.

## 2.4. Equivariant Networks

A related branch of work encoding problem structure focuses on equivariance. We say that a model $T$ is equivariant with respect to if $T(f(x)) = g(T(x))g$ (Tai et al., 2019). Where with relative position encoding we often want to be able to encode translation invariance, equivariance provides a more general framework. Equivariance has been applied to improve performance on problems with a wide array of structures, ranging from rotation-invariance (Esteves et al., 2017; 2018; Worrall et al., 2017), 3D reference frame-invariance (Liao & Smidt, 2022; Fuchs et al., 2020) and many others. The subset of these works that focus on generating equivariant token embeddings for transformers can be combined directly with LieRE or another rotation-based position encoding.

## 2.5. Lie Groups in Machine Learning

Lie groups have also had extensive use in machine learning. The range of works is diverse, ranging from algebraic signal processing (Kumar et al., 2024), automated discovery of symmetries (Forestano et al., 2023) to state estimation (Falorsi et al., 2019). Furthermore (Gallier & Quaintance, 2020) provides a friendly introduction to differential geometry and lie groups that may be of interest to the reader.

# 3. Background

## 3.1. Lie Groups in the Context of Attention

In this section, we aim to provide a minimal introduction to Lie groups so that the reader is able to understand the mathematical motivations behind LieRE. Lie groups are well studied, especially in the context of representation theory, and standard texts including (Fulton & Harris, 2013) are able to provide a more extensive introduction to the subject.

In this context, Lie groups are smooth sets of matrices that are closed under matrix multiplication and inversion. For every Lie group, the matrix exponential provides a smooth bijective map from a subset of $\mathbb{R}^{n \times n}$, also known as the Lie Algebra, to the Lie group. The exponential map is a diffeomorphism and has the following key property for $U, V \in \mathbb{R}^{n \times n}$ close together:

$$\exp(U - V) = \exp(-V + U) \approx \exp(V)^{-1}\exp(U) \quad (1)$$

Both RoPE (in the context of text) and RoPE-Mixed use block-diagonal rotation matrices with 2D rotations as blocks 3b. These form a special Lie group that is commutative,

allowing us to strengthen the statement in (1) to

$$\exp(U - V) = \exp(U)\exp(V)^{-1} = \exp(V)^{-1}\exp(U). \quad (2)$$

Our work examines the tradeoff between using the stronger property in (2) or increased capacity and the weaker property (1).

## 3.2. Attention Mechanism

LieRE is a modification of the standard attention mechanism to introduce positional information, which we review below. The modification we propose is independent of whether we use multiple heads, so we focus on single-headed attention for simplicity.

Let $X \in \mathbb{R}^{n \times d}$ be the set of input embeddings and $W_Q, W_K, W_v$ be learnable matrices. Let $Q = XW_Q, K = XW_K, V = XW_V$ be the keys, queries and values respectively. The outputs are computed as scores $= \frac{QK^\top}{\sqrt{d_k}}$, $\mathcal{W} = \text{softmax}(\text{scores})$ and final outputs $z = \mathcal{W}V$. We let $Q_i$ and $K_i$ denote the $i$th rows of $Q$ and $K$ respectively.

# 4. Method

LieRE is a simple modification to the attention mechanism that is presented in Algorithm 3a. Recall that we assume that positions are $n$-dimensional vectors, a matrix $A$ is skew-symmetric if $A^T = -A$, and that the matrix exponential of a skew-symmetric matrix, call it $A$, is always a high dimensional rotation matrix.

When encoding positions $p \in \mathbb{R}^n$, LieRE learns a skew-symmetric basis of matrices $\{A_i\}$ for $i \in [n]$. It encodes a position by writing it in this basis, $\sum_{i=0}^{n} p_i A_i$. We then map the resulting skew-symmetric matrix to a high-dimensional rotation via the matrix exponential. $R(p) = \exp\left(\sum_{i=0}^{n} p_i A_i\right)$ (Figure 1). Learning in the space of skew-symmetric matrices allows us to sidestep some of the difficulty that would come from learning on the manifold of rotation matrices (Figure 2).

LieRE uses the rotation matrix computed above to modify the keys and queries of the standard attention mechanism. LieRE's final step is to modify token $i$'s query and keys as $Q_i' = R(p_i)Q_i$ and $K_i' = R(p_i)K_i$. This modifies the score between tokens $i, j$ to be $X_i^T W_Q^T R(p_i)^T R(p_j) W_K X_j$. Recalling that $R^T = R^{-1}$ for any orthogonal matrix $R$ helps illustrate the encoding of relative positions in (1). Note that the *only* difference between LieRE and RoPE-Mixed is that the latter constrains the rotations to be block-diagonal with block size two.

We include the pseudocode for the LieRE attention in Algo-

---

**Algorithm 1** LieRE Attention

---

1: **procedure** LIERE_ROTATIONS($p, A$)
2:    $d \leftarrow dimension(p)$
3:    **return** $\mathrm{matrix\_exp}\left(\sum_{i=0}^{p} A_i p_i\right)$
4: **end procedure**

5: **procedure** LIERE_ATTENTION($Q, K, V, A$)
6:    $p \leftarrow \mathrm{tokenPositions}$
7:    $R \leftarrow$ LIERE_ROTATIONS($p, A$)
8:    *// Multiply each key and query vector by the rotation for that token.*
9:    $K_{\mathrm{rot}} \leftarrow$ BATCHMATMUL($R, K$)
10:    $Q_{\mathrm{rot}} \leftarrow$ BATCHMATMUL($R, Q$)
11:    $\mathrm{Attention} \leftarrow \mathrm{softmax}\left(\frac{Q_{\mathrm{rot}} K_{\mathrm{rot}}^T}{\sqrt{\dim(K)}}\right) V$
12:    **return** Attention
13: **end procedure**

---

(a) Lie Rotary Embedding (LieRE) attention mechanism.

---

**Algorithm 2** RoPE Attention

---

1: **procedure** ROPE($X, p, d$)
2:    $\theta \leftarrow \frac{1}{10000^{2i/d}}$   $\forall i \in [0, d)$
3:    **for** $i \leftarrow 0$ **to** $d$ **step** 2 **do**
4:      $X_{\mathrm{rot}}^{i} \leftarrow X^i \cos p\theta^i - X^{i+1} \sin p\theta^i$
5:      $X_{\mathrm{rot}}^{i+1} \leftarrow X^i \sin p\theta^i + X^{i+1} \cos p\theta^i$
6:    **end for**
7:    **return** $X_{\mathrm{rotated}}$
8: **end procedure**

9: **procedure** ROPE_ATTENTION($Q, K, V$)
10:    $p \leftarrow \mathrm{tokenPositions}$
11:    $d \leftarrow \mathrm{embeddingDimension}$
12:    $K_{\mathrm{rot}} \leftarrow$ ROPE($K, p, d$)
13:    $Q_{\mathrm{rot}} \leftarrow$ ROPE($Q, p, d$)
14:    $\mathrm{Attention} \leftarrow \mathrm{softmax}\left(\frac{Q_{\mathrm{rot}} K_{\mathrm{rot}}^T}{\sqrt{d}}\right) V$
15:    **return** Attention
16: **end procedure**

---

(b) Rotary Position Embedding (RoPE) attention mechanism.

*Figure 3.* Comparison of the LieRE and RoPE-Mixed attention mechanisms.

*Table 1.* 2D image and 3D video classification Top-1 Accuracy (95% confidence intervals) results. All models use 85.2M parameters for 2D tasks and 88.7M parameters for 3D task (Krizhevsky et al., 2009; Deng et al., 2009; Soomro et al., 2012; Flanders et al., 2020) * equivalent to DeiT architecture, ** equivalent to Vivit (spatio-temporal) architecture.

| Method | CIFAR-100 | ImageNet-1k | UCF101 |
|---|---|---|---|
| Abs. Pos. E.*,** | 63.9 (62.9-65.8) | 66.1 (65.7-66.5) | 40.9 (40.5-41.3) |
| VisionLlama RoPE | 65.5 (64.6-66.5) | 65.4 (65.0-65.8) | 45.0 (44.6-45.4) |
| RoPE-Mixed | 68.8 (67.9-69.7) | 68.8 (68.4-69.2) | 46.3 (45.9-46.7) |
| LieRE$_8$ | **70.3 (69.4-71.2)** | **69.6 (69.2-70.0)** | **47.0 (46.6-47.4)** |
| LieRE$_{64}$ | 70.0 (69.1-70.9) | 69.3 (68.9-69.7) | 44.7 (44.3-45.1) |

*Table 2.* P-values comparing RoPE-Mixed and LieRE$_8$ across datasets.

| Dataset | P-value (RoPE-Mixed vs. LieRE$_8$) |
|---|---|
| CIFAR-100 | $1.3 \times 10^{-5}$ |
| ImageNet-1k | $6.3 \times 10^{-3}$ |
| UCF101 | $7.1 \times 10^{-4}$ |
| $384 \times 384$ (Resolution Invariance) | $4.0 \times 10^{-4}$ |

## 5. Experiments

We evaluate LieRE across a range of tasks to assess its impact on transformer performance in 2D and 3D vision, spatial reasoning, and resolution generalization. Our goal is to isolate the contribution of positional encodings by using consistent architectures, training procedures, and hyperparameters across all methods. To this end, we utilize a standard recipe and build on top of the vision transformers repository (Yoshioka, 2024) and verify that our baselines perform similarly to prior work (Lee et al., 2022). We avoid using pre-trained weights in order to help reproducibility and comparability of the results between methods. We provide confidence interval estimates using bootstrap (B=1000, $\alpha = 0.05$). We evaluate two versions of LieRE, distinguished by the basis matrix block-diagonal sizes of 64 and 8, referred to as LieRE$_{64}$ and LieRE$_8$, respectively. Notably, a tile size of 2 corresponds to RoPE-Mixed.

rithm 3a in addition to the standard RoPE attention (Algorithm 3b). In practice, we compute the rotation matrices at the start of the forward pass. By default, the skew symmetric bases are learned separately for every layer and attention head except in the experimental section focused on sharing parameters across heads and layers. Adjusting the skew-symmetric basis matrices' block width allows us to incrementally adjust the capacity allocated towards position encoding. We specify the basis block width as a subscript, eg. LieRE$_8$. When not specified, the block size is equal to the head dimension. If we set the block size to 2, we recover RoPE-Mixed (Heo et al., 2024). For a ViT-B model the head dimension is 64.

## 5.1. 2D Image Classification

We begin with CIFAR-100 and ImageNet-1k benchmarks to evaluate LieRE in 2D vision tasks. All models use ViT-based architectures trained from scratch with standard data augmentations (RandAugment). We compare LieRE to absolute positional encodings, RoPE-Mixed (Heo et al., 2024), and VisionLLaMA (Chu et al., 2024). Table 1 shows that LieRE outperforms all baselines. On CIFAR-100, LieRE$_8$ achieves a statistically significant relative improvement in top-1 accuracy relatively of 10.0% over absolute encodings, 7.3% over VisionLLaMA, and 2.2% over RoPE-Mixed. Similar trends hold on ImageNet (Table 2).

We also investigate performance across model sizes (ViT-Tiny, Base, Large). As shown in Table 6 and Figure 8, LieRE$_8$ consistently outperforms other methods.

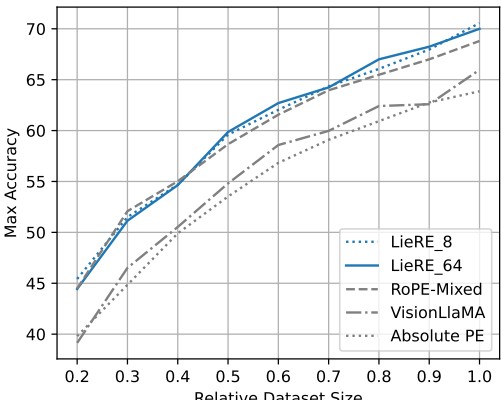

*Figure 4.* Performance comparison of different positional embedding methods across varying dataset sizes. The plot shows the peak accuracy achieved by LieRE$_8$, LieRE$_{64}$, RoPE-Mixed, VisionLlaMA, and Absolute PE when trained on different fractions of the CIFAR-100 dataset. Both LieRE variants and RoPE-Mixed consistently outperform other methods, with their advantage becoming particularly pronounced in data-scarce scenarios.

To evaluate robustness in low-data regimes, we perform a data ablation study. Figure 4 shows that LieRE variants and RoPE-Mixed maintain significantly higher accuracy than baselines when training on only 20–90% of the CIFAR-100 dataset. This highlights LieRE's data efficiency.

*Table 3.* Comparing different positional encodings on synthetic task on base model (85M)* equivalent to DeiT architecture across different resolutions, $108 \times 108$, $168 \times 168$, $276 \times 276$

| Method | 108 | 168 | 276 |
|---|---|---|---|
| Abs. Pos. E.* | 45.1 (43.1-47.7) | 41.0 (39.0-43.2) | 40.1 (38.2-42.0) |
| Abs. Pos. E.* (2M examples) | 47.2 (45.2-49.1) | - | - |
| VisionLlama RoPE | 46.4 (44.0-47.9) | 48.6 (46.0-49.9) | 48.6 (46.2-50.1) |
| RoPE-Mixed | 100 (99.5-100) | 98.6 (97.6-98.7) | 88.6 (87.4-89.9) |
| LieRE$_8$ | 99.5 (99.2-99.8) | 99.7 (99.4-99.9) | 99.7 (99.5-99.9) |
| LieRE$_{64}$ | **100 (99.5-100)** | **100 (99.6-100)** | **100 (99.7-100)** |

## 5.2. Synthetic Spatial Reasoning Task

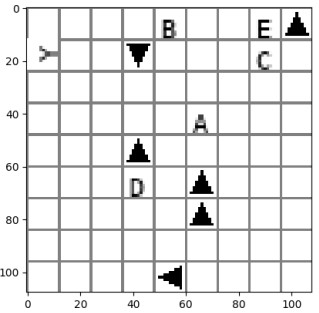

*Figure 5.* The model is asked to identify the direction of the arrow pointed to by the base of the Y. Here said arrow points down. This task requires understanding spatial relationships.

Recent observations indicate that even advanced models like ChatGPT-4 and Claude Sonnet 3.5 struggle with basic spatial reasoning tasks. To investigate whether these limitations stem from positional encoding mechanisms, we designed a synthetic image classification task (Shah et al., 2024). The task presents a $108 \times 108$ pixel image containing a $9 \times 9$ grid (81 cells). Within this grid, we randomly place eight arrows and six letters (A, B, C, D, E, and Y), ensuring that one arrow is placed in the direction of the base of the "Y". The objective is to identify the direction of this specific arrow. To introduce visual complexity, we include four spurious letters and seven additional arrows as distractors. Figure 5 illustrates an example of this task setup. We train the models on 800,000 examples and observe that they generally converge after the first 400,000 examples.

We verified that none of ChatGPT 4o, Claude Sonnet 3.5 and Gemeni Pro 1.5 are able to solve this task.

Table 3 outlines the performance of different positional encoding methods on a synthetic task using a base model (85M parameters) equivalent to the DeiT architecture. We evaluate the models across three different input resolutions ($108 \times 108$, $168 \times 168$, and $276 \times 276$ pixels), revealing differences in scalability and effectiveness. While absolute positional encoding shows degraded performance as resolution increases (from 45.1% to 40.1%) and RoPE-Mixed demonstrates strong but degrading performance at higher

*Table 4.* Accuracy with parameter sharing over heads and layers for ViT-B sized models on CIFAR-100.

| FLOP | All Shared | Shared Across Layers | Shared Across Heads | RoPE-Mixed | LieRE$_{64}$ | LieRE$_8$ |
|---|---|---|---|---|---|---|
| 5.684G | | ✓ | ✓ | 68.8 | 70.0 | **70.3** |
| 5.684G | | ✓ | | 68.7 | 69.5 | **69.8** |
| 5.613G | | | ✓ | 69.5 | 69.7 | **69.7** |
| 5.613G | ✓ | | | 68.3 | 69.4 | **69.5** |

resolutions (from 100% to 88.6%), both LieRE variants maintain near-perfect accuracy across all resolution scales, with LieRE$_{64}$ achieving 100% accuracy consistently. Qualitative analysis shows that absolute and visionllama position encoding attend less clearly to the "Y" token than RoPE-Mixed and LieRE. Please refer to the appendix for attention map examples, Figure 10 and Figure 12.

### 5.3. 3D Classification

To assess LieRE's performance on 3D data, we use the UCF101 video classification benchmark (Soomro et al., 2012). All models use a ViT-style backbone with 3D patch tokenization, trained from scratch with no hyperparameter tuning and the dataloader from (Tong et al., 2022). The full set of hyperparameters may be found in appendix B.1. We observe a relative accuracy improvement of the LieRE-based transformer of up to 15.1% compared to absolute position embeddings and at least 1.5% compared to RoPE-inspired position encodings (table 1).

### 5.4. LieRE Capacity and Parameter Efficiency

LieRE introduces minimal overhead——only 580k additional parameters (0.68% for ViT-B)——yet offers a flexible mechanism for increasing representational capacity. To explore the impact of this marginal capacity, we vary the density of the skew-symmetric basis and examine parameter sharing across heads and layers.

We control capacity via imposing a block-diagonal structure on the basis matrices. Smaller blocks (e.g., $2 \times 2$) replicate RoPE-Mixed, while larger blocks increase expressivity, with LieRE$_{64}$ using a fully dense basis. We adopt RoPE-Mixed's initialization for fair comparison. We observe that RoPE-Mixed is more sensitive to initialization than LieRE (Appendix Table 7).

Figure 6 shows performance as a function of block size. Both in 2D (CIFAR-100) and 3D (UCF101), accuracy improves with block size, peaking around $8 \times 8$——suggesting this is a sweet spot between capacity and regularization.

We also assess the effect of parameter sharing on CIFAR100 (Table 4). Learning LieRE parameters independently per head and per layer yields the best results. Sharing across layers or heads reduces accuracy, but still outperforms RoPE-

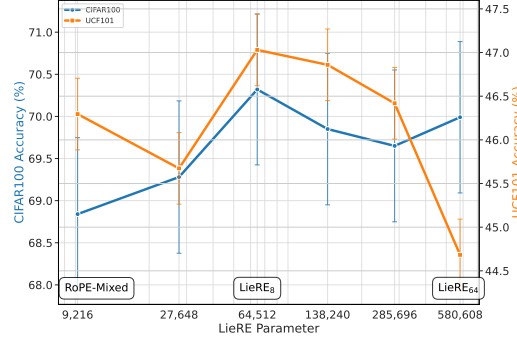

*Figure 6.* Performance varies with skew-symmetric basis learnable dimensions, referred to as LieRE parameters. This is equivalent to increasing the tile size in the skew-symmetric basis ($2 \times 2$, $4 \times 4$, $8 \times 8$, $16 \times 16$, $32 \times 32$, $48 \times 48$). For both 2D (CIFAR-100) and 3D (UCF101) LieRE with tile size $8 \times 8$ performs superior.

*Table 5.* Relative accuracy drop for 2D image classification (CIFAR-100) and Video recognition (UCF101) after patch shuffling

| Method | CIFAR-100 (2D) | | | UCF101 (3D) | | |
|---|---|---|---|---|---|---|
| | Before Shuffling↑ | After Shuffling↓ | Drop(%) ↑ | Before Shuffling↑ | After Shuffling↓ | Drop(%) ↑ |
| Abs. Pos. E. | 63.9 | 19.6 | 69.3 | 40.9 | 39.5 | 0.0 |
| VisionLlama RoPE | 65.5 | 29.7 | 54.8 | 45.0 | 37.0 | 17.7 |
| RoPE-Mixed | 68.8 | 17.1 | 75.1 | 46.3 | 28.2 | 39.1 |
| LieRE$_8$ | 70.3 | 12.3 | 82.5 | 47.0 | 27.8 | **40.9** |
| LieRE$_{64}$ | 70.0 | 10.8 | **84.6** | 44.7 | 28.0 | 37.4 |

Mixed.

### 5.5. Patch shuffling: Measuring Positional Dependency

Shuffling patches and frames allows us to see how much the model is able to use the positional information in its inputs. A model whose architecture does not allow/encourage the use of positional information will converge to a representation similar in spirit to a bag-of-words, where the relative locations of pixels/voxels do not matter. A greater drop-off in accuracy during shuffling is indicative that the model more heavily utilizes positional information.

We evaluate models using the decline in accuracy when evaluating on shuffled patches. We observe the most significant decline LieRE-based transformers, leading to the conclusion that LieRE models rely more on positional information as expected. The complete results are displayed for CIFAR-100 and table for UCF101 (table 5).

### 5.6. Multi-resolution Classification

In this section we compare the ability of methods to generalize to image resolutions not seen during training. We evaluate two training recipes inspired by (Heo et al., 2024).

The first recipe matches the rest of the paper and consists of training the models on images of size $224 \times 224$ for 200

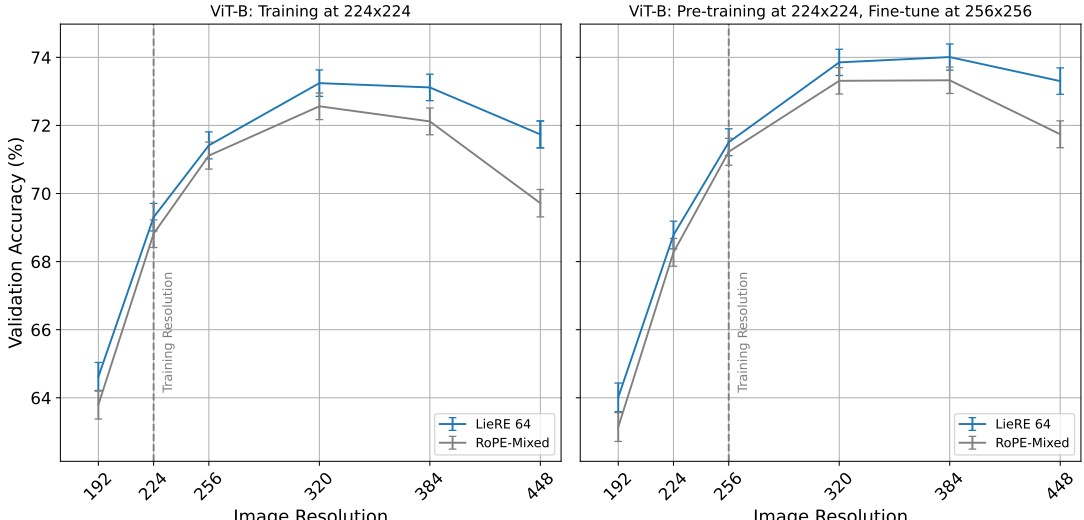

*Figure 7.* Validation accuracy comparison between LieRE 64 and RoPE-Mixed positional embeddings across different image resolutions on ImageNet. Left: Models trained at $224 \times 224$ resolution. Right: Models pre-trained at $224 \times 224$ and fine-tuned at $256 \times 256$ resolution. Both approaches show similar performance trends up to $320 \times 320$, after which LieRE 64 demonstrates significantly better accuracy retention at higher resolutions, particularly in the fine-tuned scenario.

epochs. The second adds an additional fine-tuning step at size $256 \times 256$ for 30 epochs. The full details can be found in appendix B.1.

We evaluate the accuracy on the ImageNet validation set with varying inference resolutions. Specifically, we scale the input images to resolutions of $196 \times 196$, $256 \times 256$, $320 \times 320$, $384 \times 384$, and $448 \times 448$ pixels per dimension, and present the resulting accuracies in figure 7.

For position assignment, we adopt a sequential approach where token positions are scaled proportionally to the image dimensions. For example, doubling the length of an image in each dimension doubles the range of positional indices. This method outperforms rescaling positions to a fixed range, as demonstrated by superior results for both RoPE-Mixed and LieRE across the evaluated training recipes.

## 6. Conclusion

We proposed Lie group Relative position Encodings (LieRE), a positional encoding method that modifies the attention method via dense, learned, high-dimensional rotation matrices. As compared to the more widely used block-2D rotation matrices typically used to encode positions, dense rotation matrices can encode both relative and absolute positional information and allow a greater portion of the model's learnable capacity be allocated to spatial reasoning. Experiments on 2D image classification (CIFAR-100, ImageNet-1k) and 3D video classification (UCF101) show that LieRE consistently outperforms existing positional en-

coding methods. Our analysis indicates that LieRE-based ViTs effectively leverage spatial reasoning capabilities unavailable to transformers using only absolute positional encodings. In addition to accuracy gains, LieRE offers notable data and compute efficiency. Its simplicity, flexibility, and strong capacity to learn spatial structure make it broadly applicable. With no tokenizer modifications beyond position output and no additional architectural changes, LieRE provides simple approach for controlling the amount of positional information into transformers.

## 7. Limitations

While LieRE shows promising results for 2D and 3D inputs, several limitations are worth noting. For 1D input, LieRE reduces to RoPE with learnable phases (proof in appendix A). Our method is designed to modify the inner product, making it compatible with most attention mechanisms, including standard softmax attention and linear attention. However, this may limit its applicability to other architectures——such as convolutional neural networks——that do not rely on the attention mechanism. Future work could adapt the method to a broader range of architectures. The current formulation encodes vector positions in $\mathbb{R}^d$. While sufficient for many applications, it may not directly apply to tasks that require pose encoding in $SE(3)$ (e.g., robotics). Lastly, in its current implementation, LieRE relies on the accuracy and numerical stability of the matrix exponential in PyTorch. Future work may explore more efficient and robust implementations or approximations of this operation. Despite

these limitations, we believe our approach provides valuable insight into improving model performance and reducing training costs by encoding relative positional information across various input dimensionalities.

## Impact statement

This paper presents work whose goal is to advance the field of Machine Learning. There are many potential societal consequences of our work, none which we feel must be specifically highlighted here.

## Acknowledgements

We would like to thank Maksim Maydanskiy for helping us understand Lie groups and Lie group representations. We would like to thank Aradhana Sinha for suggesting the shuffling experiment and providing early feedback on the paper. Furthermore, lucidrains suggested adding the proof of that in the 1D setting, LieRE has identical representational capacity to RoPE up to the learnable basis. Sophie Ostmeier was supported by the German Research Foundation (DFG), Walter-Benjamin fellowship (ID: 517316550). This project was supported by Google Cloud and Azure credits. This work was also supported in part by the Medical Imaging and Data Resource Center (MIDRC), which is funded by the National Institute of Biomedical Imaging and Bioengineering (NIBIB) under contract 75N92020C00021 and through the Advanced Research Projects Agency for Health (ARPA-H).

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

# A. Equivalence of RoPE and LieRE in one Dimension

Though focused on higher dimensional inputs LieRE remains compatible with 1D tasks. It turns out that in the 1D setting, LieRE has identical representational capacity to RoPE. This is not the case for higher dimensional inputs

Recall that in the 1D setting positions are scalars. The LieRE rotation is $R = \exp(tA)$ for a some learnable skew-symmetric matrix $A$. Recall that skew-symmetric matrices can be written in the form $S^T \Lambda S$ where and is orthogonal $\Lambda$ has the structure denoted below.

$$\Lambda = \begin{pmatrix} 0 & \lambda_0 & & & \\ -\lambda_0 & 0 & & & \\ & & 0 & \lambda_1 & \\ & & -\lambda_1 & 0 & \\ & & & & \ddots \end{pmatrix}$$

We can then use an identity of the matrix exponential to break down the LieRE rotation matrix.

$$R = \exp(tS^T \Lambda S) = S^T \exp(t\Lambda)S.$$

For two tokens in positions $t, t'$ we denote the embeddings for a specific attention head as $x_t, x_{t'}$. If $K, Q$ denote the corresponding key and query linear transformation matrices we can write the attention inner product with LieRE explicitly.

$$\begin{aligned} x_t K R_t^T R_{t'} Q x_{t'} &= x_t K S^T \exp(t\Lambda) S^T S \exp(t'\Lambda) S Q x_{t'} \\ &= x_t K S^T \exp(t\Lambda)^T \exp(t'\Lambda) S Q x_{t'} \\ &= x_t K S^T \exp(t\Lambda)^T \exp(t'\Lambda) S Q x_{t'} \end{aligned}$$

We let $K' = KS^T$ and $Q' = SQ$, since these matrices are all learnable we can fold the $S$ matrix into parameters of the key and query linear layers for the given head, allowing us to simplify the above expression.

$$x_t K' \exp(t\Lambda)^T \exp(t'\Lambda) Q' x_{t'}$$

Now we use the fact that each block is skew symmetric. In the case of two dimensions,

$$\exp\left(\begin{pmatrix} 0 & \lambda \\ -\lambda & 0 \end{pmatrix}\right) = \begin{pmatrix} \cos(\lambda) & \sin(\lambda) \\ -\sin(\lambda) & \cos(\lambda) \end{pmatrix}$$

If we let $R_\lambda$ denote a block diagonal rotation matrices with 2D rotations of angles $\lambda_0, \ldots, \lambda_n$, we can rewrite the above expression in a more familiar form.

$$x_t K' R_t^T R_{t'} Q' x_{t'}$$

This is exactly the formulation of the original RoPE position encoding. This also makes more clear how LieRE is different from RoPE-Mixed in the high dimensional setting. The above proof depends on the fact that we can decompose every rotation into a matrix of the form $S^T \Lambda S$ with $S$ not dependent on the position, allowing us to fold the orthogonal $S$ matrices into the key and query matrices. This decomposition with constant $S$ is guaranteed because the inputs to the matrix exponential differ by only a scalar factor. This is no longer true once we switch to more than a one-dimensional basis of skew symmetric matrices.

# B. Experimental Details

## B.1. Experimental Hyperparameters

The backbone for all experiments is configured as ViT-B, with 12 layers, a hidden dimension of 768, and an intermediate dimension of 3096. We use a dropout of 0.1. We used CLS pooling in our implementation to facilitate comparability with

existing literature in the field. Further experiments revealed substantial performance improvement with mean pooling and LieRE. We use the pytorch lightning framework for all experiments (Falcon, 2019).

### B.2. 2D Image Classification

The CIFAR experiments where trained on 8xL4 GPUs with 24GB of VRAM each and all took under 30 minutes to complete. The basis capacity scaling experiment was conducted using RTX6000 GPUs. The ImageNet experiments were trained on 8xL40 GPUs and all took less than 2 days and 5 hours of runtime including time lost due to preemption and resource sharing. We use a cosine learning rate schedule with an initial learning rate of $1E - 4$ and train for 200 epochs. We use an effective batch size of 512. We use a patch size of $4 \times 4$ on the original $32 \times 32$ image for CIFAR-100 and a patch size of $16 \times 16$ on the randomly cropped and resized $224 \times 224$ image. All vision experiments used RandAugment (Cubuk et al., 2020). We use the ADAM optimizer with betas of 0.9 and 0.999 and $\epsilon = 1e - 8$. The hyperparameters were tuned with RoPE-Mixed and selected before conducting the LieRE trainers as to ensure a fair comparison.

### B.3. 3D Video Classifications

The 3D classification experiments were conducted on either $8 \times A100$ 40GB GPUs or $4 \times A100$ 80GB GPUs with the effective batch size held constant either by using a gradient accumulation or increasing the batch size. Similar to 2D classification, we use an initial learning rate of $1E - 4$ with a cosine decay, trained for 200 epochs, and had a total batch size of 64 and a patch size of $2 \times 16 \times 16$ on the randomly cropped and resized $8 \times 224 \times 224$ video/image. We use the ADAM optimizer with betas of 0.9 and 0.999 and $\epsilon = 1e - 8$.

### B.4. Multi-resolution Classification

The second training recipe consists of 30 epochs with an initial learning rate of 1E-5 with a cosine decay. This mirrors the DEIT III training recipe that first pretrains at a lower resolution and finetunes at a higher resolution.

### B.5. CIFAR-100 Performance Across Model Scales

We further evaluate the impact of incorporating LieRE across different model sizes on CIFAR-100, as shown in Table 6. LieRE consistently outperforms the baseline with statistically significant gains across all three model scales. However, these results may be sensitive to dataset size, as all models are trained from scratch in this study. This is reflected in the performance drop observed with the ViT-Huge model.

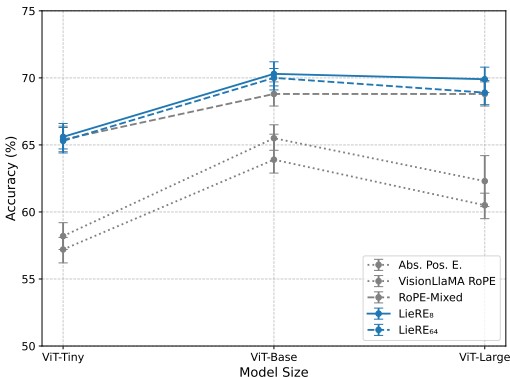

*Figure 8.* Performance behavior on CIFAR-100 (2D Image Classification) over ViT-Tiny (22M), ViT-Base (85M), ViT-Large (302M) for LieRE RoPE-Mixed and Absolute Encoding (Appendix, table 6).

### B.6. Initialization Sensitivity

In order to understand the sensitivity of the methods to the initialization, we explore several scaling factor for the weights initialization. The results are presented in table 7.

*Table 6.* Comparison of Position Encoding Methods for Different ViT Models Sizes on CIFAR-100, Accuracy (bootstrapped 95%CI)

| Position Enc. | ViT-Tiny (22M) | ViT-Base (86M) | ViT-Large (302M) |
|---|---|---|---|
| Abs. Pos. E. | 57.2 (56.2-58.1) | 63.9 (62.9-65.8) | 60.5 (59.5-61.4) |
| VisionLlaMA RoPE | 58.2 (57.2-59.2) | 65.5 (64.6 -66.5) | 62.3 (60.4-64.2) |
| RoPE-Mixed | 65.4 (64.5-66.4) | 68.8 (67.9-69.7) | 68.8 (67.9-69.7) |
| $\text{LieRE}_8$ | **65.6 (64.7-66.6)** | **70.3 (69.4-71.2)** | **69.9 (68.9-70.8)** |
| $\text{LieRE}_{64}$ | 65.3 (64.4-66.3) | 70.0 (69.1-69.7) | 68.9 (68.0-69.8) |

*Table 7.* Paired Z-test between $2\pi$ and $1$ initialization for $\text{LieRE}_8$ and RoPE-Mixed.

| Metric | $\text{LieRE}_8$ ($2\pi$ vs 1 init) | RoPE-Mixed ($2\pi$ vs 1 init) |
|---|---|---|
| Z-statistic | -1.81 | -3.85 |
| P-value | 0.070 | 0.00012 |
| Difference between means | -0.0118 | -0.0255 |
| 95% Confidence Interval | [-0.0246, 0.0010] | [-0.0385, -0.0125] |

# C. Python implementation of LieRE rotation matrix computation

```python
basis_raw_params = nn.Parameter(
    torch.rand(
        input_dimensionality,
        head_dim,
        head_dim,
    ) * 2 * math.pi # optional, inspired from RoPE-Mixed paper
)
upper_triangle = (
    torch.triu(basis_raw_params, diagonal=1)
)
skew_bases = upper_triangle - torch.transpose(upper_triangle, -1, -2)
in_basis_positions = (
    positions.reshape(list(positions.shape) + [1] * 2) * skew_bases
)
rotation_log = torch.sum(in_basis_positions, dim=-3)
rotation = torch.matrix_exp(rotation_log.to(dtype=torch.float32))
rotation = rotation.to(dtype=positions.dtype)
```

# D. Compute efficiency

We demonstrate that LieRE-based transformer achieves comparable performance to the Absolute Position Embedding baseline (DeiT III (Touvron et al., 2022)) on CIFAR-100 with fewer training epochs. This represents a notable advancement over recent methods such as VisionLlama and RoPE-Mixed (Chu et al., 2024; Heo et al., 2024). Figure 9 illustrates that LieRE enables a $3.9\times$ reduction in training compute while maintaining the accuracy achieved by absolute position encodings after 200 epochs.

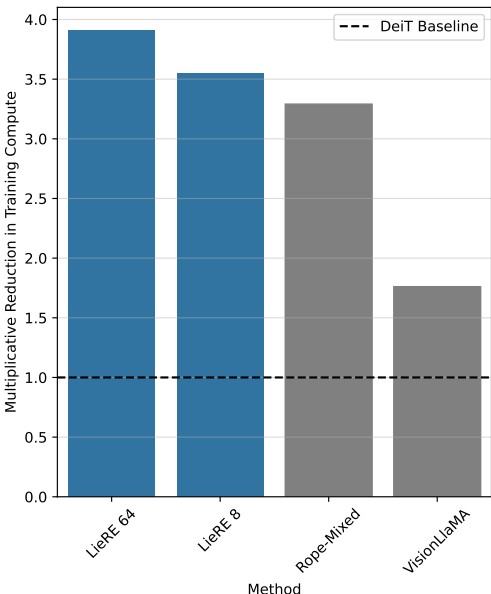

*Figure 9.* The LieRE spatial encoding allows the model to match the performance of absolute position encodings with substantially less training time.

### D.1. FLOPS Comparison of methods

We find that since all methods we examine introduce a computational cost that is at most linear in the number of tokens, and runtime is dominated by the quadratic attention component, there is no substantial difference in computational efficiency between the methods. We list inference FLOP of the various methods in table 8.

*Table 8.* FLOP analysis with percentage increase compared to absolute position encodings

| Position Enc. | ViT-Tiny (22M) | ViT-Base (85M) | ViT-Large (302M) |
| --- | --- | --- | --- |
| Abs. Pos. E.[*] | 0.963G | 5.607G | 19.856G |
| VisionLlaMA RoPE | 0.963G (+0.001%) | 5.607G (+0.002%) | 19.856G (+0.000%) |
| RoPE-Mixed | 0.964G (+0.104%) | 5.609G (+0.036%) | 19.863G (+0.035%) |
| $LieRE_8$ | 0.968G (+0.519%) | 5.617G (+0.178%) | 19.882G (+0.065%) |
| $LieRE_{64}$ | 0.970G (+0.727%) | 5.684G (+1.375%) | 20.061G (+1.033%) |

## E. Validation Losses

*Table 9.* 2D image and 3D video classification Top-1 Validation loss (95% confidence intervals) results. All models use 85.1M parameters for 2D tasks and 88.7M parameters for 3D task (Krizhevsky et al., 2009; Deng et al., 2009; Soomro et al., 2012; Flanders et al., 2020) [*] equivalent to DeiT, [**] equivalent to Vivit (spatio-temporal).

| Method | CIFAR-100 | ImageNet-1k | UCF101 |
| --- | --- | --- | --- |
| Abs. Pos. E.[*,**] | 1.56 (1.47-1.56) | 1.84 (1.81-1.86) | 2.94 (2.92-2.96) |
| VisionLlama RoPE | 1.56 (1.51-1.61) | 1.98 (1.94-2.01) | 2.66 (2.63-2.69) |
| RoPE-Mixed | 1.38 (1.33-1.43) | 1.72 (1.68-1.74) | 2.52 (2.49-2.54) |
| $LieRE_8$ | **1.36** (1.31-1.41) | **1.73** (1.70-1.76) | **2.47 (2.44-2.49)** |
| $LieRE_{64}$ | 1.37 (1.33-1.42) | 1.73 (1.70-1.76) | 2.64 (2.62-2.67) |

## F. Basis parameters scaling

*Table 10.* Accuracy Results for Different LieRE$_\Theta$ Parameters, $^*$ relative to the model size of 85.2M and 88.7M

| Dataset | LieRE$_\Theta$ Parameter Absolute | LieRE$_\Theta$ Parameter Relative $^*$ | Tile Size | Accuracy (%) | CI (95%) |
|---------|-----------|-----------|-----------|-----------|-----------|
| CIFAR100 | 9216 | 0.01 % | 2 | 68.84 | (67.93-69.75) |
| CIFAR100 | 27648 | 0.03 % | 4 | 69.28 | (68.38-70.18) |
| CIFAR100 | 64512 | 0.08 % | 8 | 70.32 | (69.42-71.22) |
| CIFAR100 | 138240 | 0.16 % | 16 | 69.85 | (68.95-70.75) |
| CIFAR100 | 285696 | 0.34 % | 32 | 69.65 | (68.75-70.55) |
| CIFAR100 | 580608 | 0.68 % | 64 | 69.99 | (69.09-70.89) |
| UCF101 | 9216 | 0.01 % | 2 | 46.30 | (45.89-46.71) |
| UCF101 | 27648 | 0.03 % | 4 | 45.67 | (45.26-46.08) |
| UCF101 | 64512 | 0.08 % | 8 | 47.03 | (46.62-47.44) |
| UCF101 | 138240 | 0.16 % | 16 | 46.86 | (46.45-47.27) |
| UCF101 | 285696 | 0.32 % | 32 | 46.42 | (46.01-46.83) |
| UCF101 | 580608 | 0.66 % | 64 | 44.68 | (44.27-45.09) |

## G. Attention Maps

We show the attention scores to the CLS token, averaged across heads for every layer. Each grid is rescaled so that the minimum element has value zero and maximum value of one. Red indicates the maximal score, and blue indicates the minimal value.

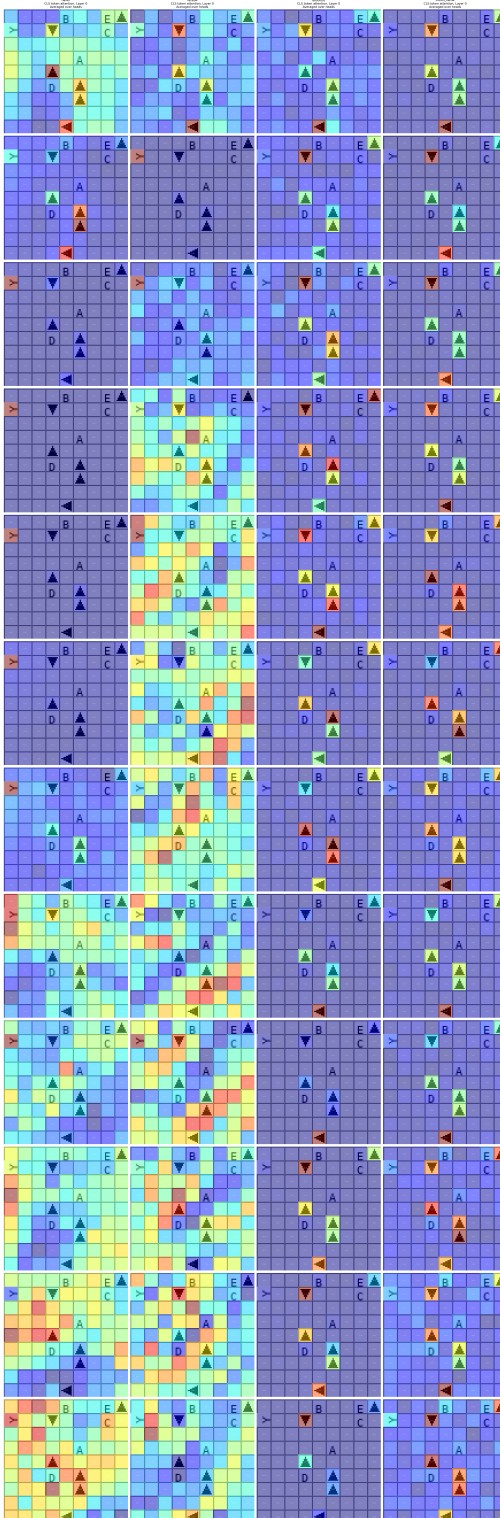

Figure 10. Attention Maps (normalized), $108 \times 108$: RoPE-Mixed, LieRE, Absolute and Visionllama positional encoding (x-axis), Layers 1-12 (y-axis). While RoPE-Mixed and LieRE learn to look at the "Y", Absolute and Visionllama less and concentrate on the arrows.

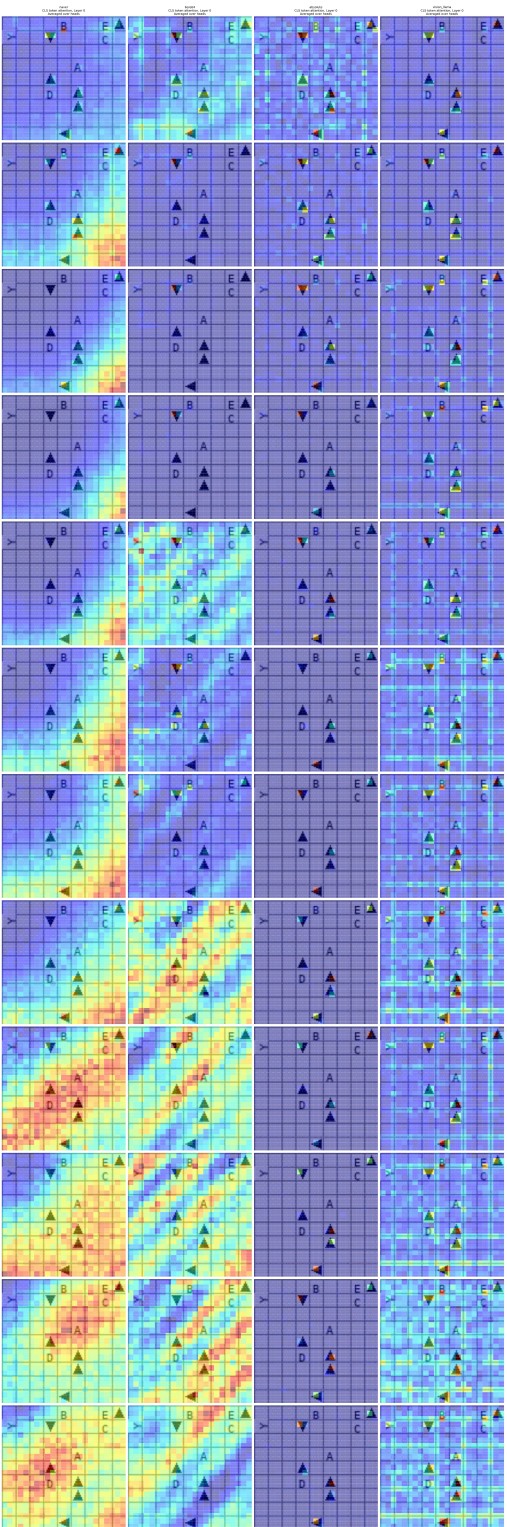

Figure 11. Attention Maps (normalized), $276 \times 276$: RoPE-Mixed, LieRE, Absolute and Visionllama positional encoding (x-axis), Layers 1-12 (y-axis). While LieRE still learns to look at the "Y", Absolute and Visionllama do less so.

Figure 12. Comparison of attention maps across different resolutions and position encoding methods.

