# OpenReview forum: "LieRE: Lie Rotational Positional Encodings"
_ICML.cc/2025/Conference — ICML 2025 poster_

### Official Review · Reviewer_yH2U · 2025-03-07

**Overall Recommendation:** 3

**Summary:**

The authors introduce a type of positional embedding which extends the RoPE embeddings by introducing learnable rotation matrices.

## update after rebuttal
I thank the authors for their thorough response. In light of this, I will increase my score to weak accept.

**Claims And Evidence:**

The authors present reasonable although somewhat limited experimental validation, by training ViT models on CIFAR-10, Imagenet-1k and UCF101.
The gap between their model and RoPE mixed (from which it is an incremental modification) is very small.

**Essential References Not Discussed:**

N/A

**Experimental Designs Or Analyses:**

N/A

**Methods And Evaluation Criteria:**

Yes

**Other Comments Or Suggestions:**

None

**Other Strengths And Weaknesses:**

Strengths:
- Paper is well-written and easy to follow
- Method seems to marginally outperform existing methods at little compute increase

Weaknesses:
- Novelty: as acknowledged by the authors, this method is an incremental modification of the existing RoPE-Mixed embeddings where instead of having block matrices of block size 2, the block size becomes a hyperparameter.
- Effectiveness: I am not convinced of the benefits of this method. First, as shown in figure 5 and table 2, the increase in performance is rather marginal. Second, as shown in Figure 8, the effect of this hyperparameter on performance is rather unpredictable, which does not make this method particularly practical. Although the sections 5.2 and 5.6 of the paper are a bit more convincing, I remain lukewarm about the effectiveness of the method.
- Unpolished: the paper seems to have been rushed nearing the deadline and feels unpolished. Consider section 5.2: the second paragraph ends with an unfinished sentence ("The only exception to this is absolute position encodings, where we have variants trained on 800,000 and") and the third paragraph contains an undefined reference. Additionally, figures 1 and 2 are not very clear in my opinion.

**Questions For Authors:**

None

**Relation To Broader Scientific Literature:**

The litterature review is thorough, and I appreciate that the authors are honest in acknowledging similitudes with existing methods ("Note that the only difference between LieRE and RoPE-Mixed is that the latter constrains the rotations to be block-diagonal with block size two").

**Theoretical Claims:**

N/A

---

> ### Author Rebuttal · Authors · 2025-04-01
>
> Dear Reviewer yH2U,
>
> Thank you for your thoughtful review and detailed feedback. We understand your concerns about the incremental nature and effectiveness of our work, and would like to address these directly:
>
> **On novelty and the primary contribution of the work**: We split up our contributions broadly into (1) analysis and (2) method.
>
> **Analysis**: The closest work to ours is RoPE-Mixed. We build on their work with both a substantially larger performance delta and more extensive analysis. We aim to include coverage of technical details not present in the most similar prior work such as the sensitivity to weights initialization (see the table in the response to reviewer gAmX) and different methods of defining extrapolated token positions. In addition to an extended quantitative comparison, we aim to provide qualitative insights to how the position encodings affect inference and training dynamics with the attention maps and patch shuffling experiments. Finally, we extend experimental coverage to study the effect of the dimensionality of the input, a first even for existing position encodings. Concretely, to the extent of our knowledge, this is also the first work that benchmarks RoPE-Mixed for 3D data.
>
> **Method**: We believe strongly that the presentation of a technical work should be as easy to understand. This motivates us to keep the connection to existing methods simple, including being direct about the settings where the methods are identical and the additional complexity of LieRE is worth the additional complexity.
> Novel technical machinery is required in order to utilize high dimensional rotation matrices that are dense or have block size more than two. In particular, we introduce the method of encoding the positions in a basis of skew symmetric matrices which is then passed through the matrix exponential to obtain dense high-dimensional rotations. This use of Lie group theory is novel, and the key ingredient in enabling the use of dense high-dimensional rotations with allows LieRE to go beyond RoPE-Mixed with statistically significant performance improvements.
> | Dataset | p-value (Rope-Mixed vs. LieRE_8) |
> |---------|----------------------------------|
> | CIFAR-100 | 1.3e-05 |
> | ImageNet-1k | 6.3e-03 |
> | UCF101 | 7.1e-04 |
> | 384 x 384 (Resolution Invariance) | 4.0e-04 |
>
> Your observation about block size unpredictability helped us recognize the need to better articulate our findings: LieRE_8 consistently demonstrates optimal performance across both 2D and 3D experiments (Appendix B.9. Basis parameters, Table 8). We will incorporate clear guidelines for practitioners based on our systematic analysis.
>
> **Writing Improvements**: We will address the writing issues you identified: complete the unfinished sentence in section 5.2, fix the undefined reference and clarify Figures 1-2 with improved visual explanations (Figure 5: https://postimg.cc/D8w2tcKp).
> We believe these revisions will better communicate both the theoretical contributions and practical benefits of our work.
>
> We again thank Reviewer yH2U for their review of our paper. We hope that the above responses adequately address all concerns.
>
>
> **=== LieRE vs. RoPE 1D proof continued ===**
>
> $$
> \\begin{align*}
> x_t K R_t^T R_{t'} Q x_{t'} &= x_t K S^T \\exp(t \\Lambda) S S^T \\exp(t' \Lambda) S Q x_{t'} \\\\
> &= x_t K S^T \\exp(t \\Lambda)^T \\exp(t' \\Lambda) S Q x_{t'} \\\\
> &= x_t K S^T \\exp(t \\Lambda)^T \\exp(t' \\Lambda) S Q x_{t'}
> \\end{align*}
> $$
>
> We let $K’=K S^T$ and $Q’= S Q $, since these matrices are all learnable we can fold the S matrix into parameters of the key and query linear layers for the given head, allowing us to simplify the above expression.
> $$ x_t K’ \\exp(t \\Lambda)^T \\exp(t’ \\Lambda) Q’ x_{t’} $$
>
> Now we use the fact that each block is skew symmetric. In the case of two dimensions,
>
> $$\\exp\\left(\\begin{pmatrix} 0 & \\lambda \\\\ -\\lambda & 0 \\end{pmatrix}\\right) = \\begin{pmatrix} \\cos(\\lambda) & \\sin(\\lambda) \\\\ -\\sin(\\lambda) & \\cos(\\lambda) \\end{pmatrix} $$
>
> If we let $R_\\lambda$ denote a block diagonal rotation matrices with 2D rotations of angles $\\lambda_0, \\ldots, \\lambda_n$, we can rewrite the above expression in a more familiar form.
>
> $$ x_t K’ R_{t \\Lambda} ^T R_{t’ \\Lambda} Q’ x_{t’} $$
>
> This is exactly the formulation of the original RoPE position encoding. This also makes more clear how LieRE is different from RoPE-Mixed in the high dimensional setting. The above proof depends on the fact that we can decompose every rotation into a matrix of the form $S^T \\Lambda S$ with S not dependent on the position, allowing us to fold the orthogonal S matrices into the key and query matrices. This decomposition with constant S is guaranteed because the inputs to the matrix exponential differ by only a scalar factor. This is no longer true once we switch to more than a one dimensional basis of skew symmetric matrices.

---

### Official Review · Reviewer_gAmX · 2025-03-13

**Overall Recommendation:** 5

**Summary:**

LieRE extends the popular RoPE by replacing its block-diagonal 2D rotation matrices with learned, dense, high-dimensional rotation matrices derived from Lie group theory.
The authors show that LieRE addresses key limitations of RoPE, particularly for multi-dimensional data like images and videos. Specifically, while RoPE was originally designed for one-dimensional sequence processing (like text), LieRE generalizes position encoding to higher dimensions through the use of Lie groups. The method involves learning skew-symmetric basis matrices and computing rotation matrices for n-dimensional positions, which are then applied to keys and queries in the attention mechanism.
The paper evaluates LieRE against other positional encoding methods on several tasks, the results show that LieRE outperforms competing methods across these tasks, with particular advantages in data efficiency, resolution invariance, and when processing limited training data.

**Claims And Evidence:**

The claims made in the paper are well-supported by empirical evidence, but i just want to point out that the assertion that LieRE provides a "unified approach" for handling different dimensionalities is supported by experiments on 2d and 3d data, but testing on additional dimensionalities (like 1d sequences, higher-dimensional data) would make this claim more robust.

**Essential References Not Discussed:**

N/A

**Experimental Designs Or Analyses:**

The experimental design is sound.

**Methods And Evaluation Criteria:**

The methodology and evaluation criteria seem well designed, even though the paper focuses on classification tasks, which may not fully showcase the advantages of better positional encodings. Tasks requiring fine-grained spatial understanding (like segmentation or object detection) would provide a more comprehensive evaluation.

**Other Comments Or Suggestions:**

N/A

**Other Strengths And Weaknesses:**

Other weaknesses:
- The authors note that "RoPE-Mixed is sensitive to the initialization of weights," suggesting that LieRE might share this sensitivity, but they don't thoroughly explore how different initialization strategies might affect performance.

Other strenghts:
- The paper provides a solid mathematical foundation based on Lie group theory, extending positional encodings in beyond the current state-of-the-art methods.
- LieRE shows very good ability to generalize to image resolutions not seen during training, outperforming other methods especially at higher resolutions.
- The patch shuffling experiments offer valuable insights into how much the model utilizes positional information, with LieRE showing the most significant performance drop when positional information is disrupted.

**Questions For Authors:**

N/A

**Relation To Broader Scientific Literature:**

The paper builds directly upon RoPE and its variants (RoPE-Mixed, VisionLlama), clearly identifying limitations and proposing extensions. The application of Lie group theory to positional encodings is a new connection between abstract algebra and deep learning architectures.

The paper doesn't extensively discuss connections to other approaches for handling multi-dimensional data in transformers, such as axial attention or perceiver architectures.

Overall, while the paper makes a significant contribution to positional encoding research.

**Theoretical Claims:**

There's no formal proof that LieRE can handle "exponentially many relative positions for n-dimensional data" better than alternatives, though the empirical results are supportive.

---

> ### Author Rebuttal · Authors · 2025-04-01
>
> Dear Reviewer gAmX,
>
> Thank you for your thorough and supportive review. We particularly appreciate your recognition of our mathematical foundations and empirical results. We have built upon your feedback to further improve the paper. In addition to the changes below, we have also expanded the paper to better relate our work to other architectural work such as perceivers and axial attention.
>
> **On sensitivity to initialization**: We originally discovered the sensitivity as we were iterating to reproduce the results of RoPE-Mixed. RoPE-Mixed seems to be particularly sensitive to the scale of the initial weights. LieRE is about half as sensitive. We have added a section to the appendix that explores this in greater detail, but in short, RoPE-Mixed shows up to a 2% drop in CIFAR100 performance while  LieRE shows a 1% drop. Outside of the new initialization sensitivity experiment, all experiments for both LieRE and RoPE-Mixed are performed with the setting where RoPE mixed performs best. The key table is presented below.
> | Metric | LieRE_8 (2π vs 1 init) | RoPE-Mixed (2π vs 1 init) |
> |---------|------------------------|---------------------------|
> | Z-statistic | -1.81 | -3.85 |
> | P-value | 0.070 | 0.00012 |
> | Difference between means | -0.0118 | -0.0255 |
> | 95% Confidence Interval | [-0.0246, 0.0010] | [-0.0385, -0.0125] |
>
> **1D evaluation tasks**: We have added a section characterizing the use of LieRE for 1D tasks. In short, LieRE is equivalent in representational capacity to RoPE with learnable frequencies for one-dimensional tasks. Please see the response to reviewer “udrb” for the proof of this fact that will be included in the paper. Our empirical experiment confirms this equivalence.
> Higher-dimensional evaluation tasks: Thank you for the suggestion! To our knowledge, LieRE is the first approach to extend rotational position encodings to 3D data, and we are eager to explore scaling to 4D. Given the dataset size required to train transformers on high-dimensional data, such extensions may currently rely more on synthetic data or alternative ways of structuring dimensional information. We see this as an exciting next step and appreciate your insights on pushing this further.
>
> **Related Work**: Thank you for pointing out the connection to Axial attention and Perceivers. We have added a section of related work focused on related works towards compute-efficient scaling beyond sequence data.
> Axial Attention [19] reduces computational complexity by applying attention along specific axes (e.g., rows and columns in images), enabling transformers to scale efficiently with high-dimensional data. Perceiver [20] utilizes latent tokens to compress high-dimensional inputs into a smaller set, improving scalability as input size and dimensionality grow. These methods address the inefficiencies of traditional transformers when applied to high-dimensional data. Additionally, techniques like Swin [21] and Vmamba [22] optimize compute for visual data. Swin Transformer introduces a hierarchical approach with shifted windows, limiting attention to local regions to reduce complexity while capturing global context.  Vmamba, on the other hand, proposes a visual state space model that represents images as a collection of spatial states, allowing attention to be applied efficiently across large-scale visual inputs by exploiting spatial locality and reducing redundant computation.
> It would be great to compound these methods with LieRE, as in this work we use a plain encoder transformer.
>
> Thank you for constructive feedback which we were able to use to further improve the paper. We are excited to share this work with the community.
>
>
> **=== LieRE vs. RoPE 1D proof  ===**
>
> Though focused on higher dimensional inputs LieRE remains compatible with 1D tasks. It turns out that in the 1D setting, LieRE has identical representational capacity to RoPE. This is not the case for higher dimensional inputs for reasons that will be clearer later in the exposition. We include a cut-down version of the proof we propose to add to the paper below.
>
> Recall that in the 1D setting positions are scalars. The LieRE rotation is $R=\\exp(tA)$ for some learnable skew-symmetric matrix A. Recall that skew-symmetric matrices can be written in the form $S^T \\Lambda S$ where $S$ and is orthogonal and $$
> \\Lambda = \\begin{pmatrix}
> 0 & \\lambda_0 & & & \\\\
> -\\lambda_0 & 0 & & & \\\\
> & & 0 & \\lambda_1 & \\\\
> & & -\\lambda_1 & 0 & \\\\
> & & & & \\ddots
> \\end{pmatrix}
> $$
>
> We can then use an identity of the matrix exponential to break down the LieRE rotation matrix.
> $R = exp(tS^T \\Lambda S) = S^T \\exp(t \\Lambda ) S$. For two tokens in positions $t, t’$ we denote the embeddings for a specific attention head as $x_t,x_{t’}$. If $K, Q$ denote the corresponding key and query linear transformation matrices we can write the attention inner product  with LieRE explicitly.
>
> **... continued at the end of next rebuttal ...**

---

> > ### Comment · Reviewer_gAmX · 2025-04-06
> >
> > Thank you, I think that with these clarifications your paper will be even more solid.

---

### Official Review · Reviewer_udrb · 2025-03-14

**Overall Recommendation:** 3

**Summary:**

The authors mainly proposed a new positional encoding method called Lie, to replace the previous wildly used RoPE. It is used to improve the spatial relationship representation, especially in 2D and 3D images. Extensive experiments are conducted on classification tasks, and with the proposed PE, the accuracy values are all improved by a significant margin.

## update after rebuttal
I carefully read the authors' rebuttal, and thanks so much for the responses.
The authors also agree that it currently lacks evidences on image generation tasks, and some of the experiments are limited in design and scope. There issues are not fully resolved actually, and the authors did not clarify how to address them in the final version. The authors mainly used these experiment to "inspire" other future works, which I feel not very informative.
However, the generalization design of RoPE to Lie group itself is interesting.
So better AC can make the final decision to balance these factors.

**Claims And Evidence:**

The evidences are mostly well supporting the claims. The proposed LiePE greatly improves the transformer-based classification model by a large margin. Figure 9 also shows great generalization capability of the positional encoding to higher resolution. Experiments also show that the compute increment is not significant.

**Essential References Not Discussed:**

Not found.

**Experimental Designs Or Analyses:**

The main concerns for the experimental analysis is, the newly proposed PE is only used for image classification task and lower-res images. It is not sufficient to prove the effectiveness and expressiveness of the new PE for very long-context modeling. The image understanding tasks on the synthetic data is also very limited to prove its effectiveness. The patch shuffle experiments are interesting, but with random patch shuffle, it's not that meaningful to compare the dropping rate when the accuracy from different approaches are similarly low.

**Methods And Evaluation Criteria:**

The idea and the theory behind the proposal is elegant and interesting. Using the exponential of skew-symmetric matrix to generalize RoPE  is intuitive and smart, making the position encoding fully learnable and more expressive. Experiments on classification is a simple yet effective say of validating the idea, and the baseline comparison is clear and fair.

Given now RoPE is more verified in image generation task, it will be better to show the potential of LiePE on image generation task.

**Other Comments Or Suggestions:**

The overall writing is not that easy to follow.
Missing figure numbers in line 307, and the formatting needs improvement.
Figure5. is too hard to parse.

**Other Strengths And Weaknesses:**

See above.

**Questions For Authors:**

See above.

**Relation To Broader Scientific Literature:**

The proposed PE is supposed to be a very general approach and a plug-and-play components for all transformer-based models. The idea has its merits and it has great potential to be generalized to high-resolution image generation task. However, the experiments in this paper cannot well support the claims and might not bring significant impacts in the literature.

**Theoretical Claims:**

Not applied to this paper.

---

> ### Author Rebuttal · Authors · 2025-04-01
>
> Dear Reviewer udrb,
>
> We appreciate your recognition of our work's theoretical merits and experimental contributions. We have carefully considered your feedback and would like to address each point.
>
> **Long Context for 1D**: LieRE is primarily focused on inputs with dimensionality greater than one. In fact, there is the [non-obvious] property that, in the 1D setting, LieRE has equivalent representational ability to RoPE with learnable frequencies. This is not the case in higher dimensions. We include a proof of this fact at the end of the response, and will include it in the paper as it has been requested by other readers.
>
> **Long Context for 2D and 3D**: This implies that the natural equivalent of long-context modeling is evaluating the model at resolutions higher than it was trained at. This is the focus of section 5.6. (Multi-Resolution Classification). We evaluate with up to four times as many inference tokens than during pretraining and finetuning (Figure 9).
> Higher Resolution image generation: We agree that high resolution image generation is an exciting application to benchmark position encodings. The long-context image classification examples were motivated in part to create apples to apples experiments when compared to prior works [1, 2], as these are more focused on image classification. We are very supportive of future work in that application. This application is especially exciting in light of recent autoregressive image generation techniques such as VAR [3] that scale more predictably, enabling smaller scale experiments. It would be particularly interesting to see how position encodings could influence global consistency.
>
> **Patch shuffling**: We hope to clarify that the intent of these experiments is to provide insight to the mechanics of why the methods perform differently rather than to show one method is better than the other. Patch shuffling allows us to see whether the model is actually using positional information. We agree with you that it does not identify which method is best at the ultimate applications. For that we depend on the other experiments. The focus of the experiment is to help the reader build an intuition of what is going on with various position encodings.
> Each method compared is distinct in both how and what kind of positional information it is capable of encoding. LieRE has the ability to use both relative and absolute positional information, and, ideally, we would like to rule out there being a simpler method that could perform just as well. The drop in performance when shuffling patches is one limited datapoint consistent with that hypothesis.
>
> **Synthetic task**: We wholeheartedly agree with the assessment that the synthetic task has limitations. Still, we can see clear failures of basic spatial reasoning even in frontier models trained with resources well beyond the reach of any academic lab. Reproducing similar patterns in simple experiments greatly improves the accessibility of studying these issuesIn fact, understanding these failures is an increasingly growing area of research [6,7]. We believe better position encodings are one of the ingredients that will be necessary to resolve these limitations and hope the data point in our paper is suggestive of that. It is important to note that many VLMs are still trained with absolute position encodings that perform substantially worse than the relative position encoding we benchmarked (Table 1). While paper coherence and practical considerations prevent us from fusing this work with a new research project focused on resolving these limitations on spatial reasoning, we hope this experiment serves as a datapoint used to inform future work on spatial reasoning.
> Writing and figures: Thank you for the feedback on the writing and figures. We have used it to improve the readability of the paper and make changes such as switching to more readable high contrast color schemes for the figures. (such as Figure 5: https://postimg.cc/D8w2tcKp)
>
> Thank you for the thoughtful review, and we hope we addressed your remaining concerns.

---

### Official Review · Reviewer_SwFq · 2025-03-24

**Overall Recommendation:** 3

**Summary:**

The paper introduces a positional embedding encoding based on Lie Groups. The idea of the paper is to parameterize the positional embeddings using skew symmetric matrices. The authors show the benefit of the proposed method in terms of generalization, data efficiency and compute needed.

Overall, the idea is novel and interesting. The authors have empirical evidence that validates the quality of their work. However, the paper quality needs to be improved, both in terms of presentation as well as writing.

**Claims And Evidence:**

The paper claims are well supported by the experiments.

**Essential References Not Discussed:**

The paper overlooks several works in equivariant transformers and neural networks:

- Equivariant transformer networks, Tai et al
- Equivariant Neural Functional Networks for Transformers, Tran et al
- Equiformer: Equivariant graph attention transformer for 3d atomistic graphs, Liao et al
- Polar transformer networks, Esteves et al
- Efficient equivariant network, He et al,
- Se (3)-transformers: 3d roto-translation equivariant attention networks, Fuchs et al
- Learning so (3) equivariant representations with spherical cnns, Esteves et al

Also a lot of work in equivariance:
- Harmonic networks: Deep translation and rotation equivariance, Worral et al

Even in lie groups ML, there are plenty of references missing:
- Lie group algebra convolutional filters, Kumar et al.
- Deep learning symmetries and their Lie groups, algebras, and subalgebras from first principles, Forestano, et al
- Differential geometry and lie groups, Gallier et al
- Reparameterizing distributions on lie groups, Falorsi et al

**Ethical Review Concerns:**

None.

**Experimental Designs Or Analyses:**

Yes.

**Methods And Evaluation Criteria:**

The proposed method and evaluation criteria make sense.

**Other Comments Or Suggestions:**

See before.

**Other Strengths And Weaknesses:**

Comments:

C1  - Use a definition for the property of equations (1) and (2). For example, for eq 2 use
Comm(U,V).

C2 - Page 4, line 216 - Algorithm should be capitalized. All the subsequent calls to the word Algorithm should also be capitalized.

C3 - Page 5, a lot of blank space. I believe it would improve the quality of presentation to fix this.

C4 - The text under Figure 1 is confusing and needs a better structure. Some quantities are define but never used.

C5 - In the experiments, a piece of the text was removed, making the sentence incomplete:

“We train the models on 800,000 examples and observe that they generally converge after the first 400,000. The only exception to this is absolute position encodings, where we have variants trained on 800,000 and”

There is also a missing reference:
“Please refer to the appendix for attention map examples, Figure 13 and Figure ??.”

C6 - In page 15, the text covers the page number.

**Questions For Authors:**

The authors do not include any equivariant transformer architecture and only compare with 2 other embedding works. Why are the authors not comparing themselves with any architecture of the ones mentioned before.

I believe that the authors need to compare their results with some equivariant / quasi equivariant work of the ones mentioned before.

**Relation To Broader Scientific Literature:**

The paper expands the equivariant work in terms of positional embeddings for Lie groups.

**Theoretical Claims:**

There are not theoretical claims.

---

> ### Author Rebuttal · Authors · 2025-04-01
>
> Thank you for the thoughtful and thorough review and writing feedback which has helped strengthen the paper. We have addressed the typos and writing style in the revision based on your comments.
>
> **Equivariant work comparison**: We are excited about the equivariant line of work! We believe it is key to learning sample efficient representations for domains with extensive symmetry. We will include a new section of the related work that relates LieRE to the suggested works on equivariance and other work on lie groups in ML.
>
> As a complimentary architectural change, equivariance transformers do not directly compete with position encodings such as LieRE. Position encodings such as LieRE, RoPE-Mixed and absolute position encodings are minimally invasive modifications to the base transformer architecture, making them compatible with many different architectures. In the case of LieRE, this enables things like finetuning existing LLM weights to become multimodal models capable of handling high dimensional data.
>
> Combining them would be an exciting area of research. One of the observations present in both this work and the most closely related work [1] was that, sometimes, translation-invariance is helpful, but in other cases having access to the reference coordinate system can help performance. This is supported by the fact that adding absolute position encodings to RoPE-Mixed, and LieRE performs best in the regime where the attention patterns are not necessarily constrained to be translation invariant (recall that the attention patterns are translation invariant only when the block size is constrained to two). Fortunately, this is compatible with many of the designs in the equivariant line of work.
>
> The natural question is how do we let models benefit from both the sample efficiency of equivariance and the fact that sometimes the absolute coordinate system does contain useful information. Combining these methods in the right way is an exciting future direction but has enough technical complexity that it is hard to incorporate into the current paper without losing focus and diluting individual learnings around positional encodings. The contributions in this paper are easiest to understand when comparing to earlier works that modify the same aspect of the transformer architecture. We hope you agree that the revisions to the paper provide a stronger connection to equivariant line of work and an extended related work accelerates future research in the area.
>
> **Equivariance Related work**:A related branch of work encoding problem structure focuses on equivariance. We say that a model T is equivariant with respect to $f$ if T(f(x)) = g(T(x))$ for some $g$ [8]. Where with relative position encoding we often want to be able to encode translation invariance, equivariance provides a more general framework. Equivariance has been applied to improve performance on problems with a wide array of structures, ranging from rotation-invariance [10,13,14], 3D reference frame-invariance [9,12] and many others. The subset of these works that focus on generating equivariant token embeddings for transformers can be combined directly with LieRE or another rotation-based position encoding.
>
> **Lie Groups in Machine Learning**:Lie groups have also had extensive use in machine learning. The range of works is diverse, ranging from algebraic signal processing [15], automated discovery of symmetries [16] to state estimation [18]. Furthermore [17] provides a friendly introduction to differential geometry and lie groups that may be of interest to the reader.
>
> We thank the reviewer for highlighting this connection and helping us improve the paper.
>
> References are replaced with links to respect character limits.
> [1] https://arxiv.org/abs/2403.13298
>
> [2] https://arxiv.org/abs/2403.00522
>
> [3] https://arxiv.org/abs/2404.02905
>
> [4] https://arxiv.org/abs/2212.09748
>
> [5] https://arxiv.org/abs/2112.10752
>
> [6] https://arxiv.org/abs/2411.04097
>
> [7] https://arxiv.org/abs/2406.15955
>
> [8] https://arxiv.org/abs/1901.11399
>
> [9] https://arxiv.org/abs/2206.11990
>
> [10] https://arxiv.org/abs/1709.01889
>
> [11]https://proceedings.neurips.cc/paper_files/paper/2021/file/2a79ea27c279e471f4d180b08d62b00a-Paper.pdf
>
> [12] https://arxiv.org/abs/2006.10503
>
> [13] https://arxiv.org/abs/1711.06721
>
> [14] https://arxiv.org/abs/1612.04642
>
> [15] https://arxiv.org/abs/2305.04431
>
> [16] https://arxiv.org/abs/2301.05638
>
> [17] https://link.springer.com/book/10.1007/978-3-030-46040-2
>
> [18] https://arxiv.org/abs/1903.02958
>
> [19] https://arxiv.org/abs/1912.12180
>
> [20] https://arxiv.org/abs/2103.03206
>
> [21] https://arxiv.org/abs/2103.14030
>
> [22] https://arxiv.org/abs/2401.10166

---

> > ### Comment · Reviewer_SwFq · 2025-04-03
> >
> > I thank the authors for their response.
> > I have updated my score accordingly

---

### Decision · Program_Chairs · 2025-05-01

**Decision:**

Accept (poster)

**Comment:**

The paper had initial mixed reviews, where the major concerns were about limited experimental validation, unconvincing results, and missing discussion of related work. On the other hand, reviewers agreed that the idea is elegant and interesting and the method is built on a strong mathematical foundation.

The negative reviewers increased their score after the rebuttal so the paper ended up with unanimous acceptance recommendation. I agree with the reviewers. Although submission would be stronger with more convincing experimental results, the idea of using skew-symmetric matrices to generalize RoPE is quite interesting and I'm glad to see it being implemented.